# Effect of Microclimate on the Mass Emergence of *Hypothenemus hampei* in Coffee Grown under Shade of Trees and in Full Sun Exposure

**DOI:** 10.3390/insects15020124

**Published:** 2024-02-09

**Authors:** Valentina García-Méndez, Rebeca González-Gómez, Jorge Toledo, Javier Francisco Valle-Mora, Juan F. Barrera

**Affiliations:** 1Department of Arthropod Ecology and Pest Management, El Colegio de la Frontera Sur, Carretera Antiguo Aeropuerto km 2.5, Tapachula 30700, Chiapas, Mexico; valentina.garcia@posgrado.ecosur.mx (V.G.-M.); rgonzalez@ecosur.mx (R.G.-G.); jtoledo@ecosur.mx (J.T.); 2Researchers Program for Mexico, Consejo Nacional de Humanidades, Ciencias y Tecnologías, Av. Insurgentes Sur 1582, Col. Crédito Constructor, Benito Juárez, Mexico City 03940, Mexico; 3Statistics Area, El Colegio de la Frontera Sur, Carretera Antiguo Aeropuerto km 2.5, Tapachula 30700, Chiapas, Mexico; jav.valle@tapachula.tecnm.mx

**Keywords:** microclimate, shaded coffee plantation, *Coffea canephora*, mass emergence of insects, dispersal flight, seasonal and diel flight pattern

## Abstract

**Simple Summary:**

Rain acts as a triggering factor for the mass emergence of females of the coffee berry borer, *Hypothenemus hampei* (CBB), which shelter in old and unharvested berries, mainly in regions with a dry season. As they emerge from the old berries, thousands of CBB females fly out to search for and infest the coffee berries of the new crop. Very little information is available, however, about the effect microclimate on CBB or, in particular, about the interaction between the shade trees, the microclimate, and the mass emergence of CBB females. We conducted a field experiment in two Robusta coffee plots, one grown under shade trees and the other in full sun exposure. During the coffee intercropping period, when mass emergence of the CBB occurs, we set up a meteorological station to record microclimate data and installed traps to capture the females that fly out during the day, collecting caught individuals every hour. We found that female emergence increased after midday. Although more flying females were captured in the coffee plot grown under shade trees, the relationship between microclimatic variables and CBB mass emergence was similar for both shade conditions. This behavior has important implications for monitoring and pest management, because female CBB populations are more concentrated and exposed to natural and artificial mortality factors.

**Abstract:**

The rainfall regime has a significant impact on the microclimate and mass emergence of the coffee berry borer, *Hypothenemus hampei* (Coleoptera: Curculionidae) (CBB). Little is known, however, about the shade tree–microclimate–CBB mass emergence interaction. The objective of the present study was to compare the effect of microclimate on the mass emergence of CBB in a full sun-exposed plot with a plot shaded by trees. The experiment was conducted on a Robusta coffee farm in southern Chiapas, Mexico. In each plot, 18 traps baited with an alcohol mixture were installed to capture flying females, collecting caught individuals every hour from 8:00 to 18:00 h. A meteorological station recorded several microclimatic variables on 13 weekly sampling dates from February to May 2022. Significantly more CBB females were captured in the shaded plot. The largest number of CBB captures was recorded between 14:00 and 16:00 h for the shade plot and between 15:00 and 17:00 h for the sun-exposed plot. The mass emergence of CBB showed a positive association with precipitation, dew point, and wind speed samples and a negative association with maximum air temperature, average relative humidity, ultraviolet radiation, wind speed, and equilibrium moisture content. Our observations show that the relationship between shade trees, microclimate, and mass emergence of CBB is complex and that its study helps us to gain deeper insight into CBB bioecology and advance control techniques against this important pest.

## 1. Introduction

In the year 2020, Mexico ranked ninth among the world’s coffee-producing countries, producing four million 60 kg bags [1] on 710,405 hectares of land [2]. For Mexico, coffee production is an important source of foreign currency. Furthermore, coffee production has a social impact, providing a sustainable income for thousands of smallholder families, and contributes to numerous environmental services [3].

In Mexico, several problems affect coffee production and commercialization, such as low harvest yields, variable climate, elevated production costs, low coffee bean prices, and prevalence of pests and diseases, among others [4,5]. The coffee berry borer insect-pest *Hypothenemus hampei* (Ferrari) (Coleoptera: Curculionidae: Scolytinae) (CBB) inflicts substantial damage, causing a reduction in both the yield and quality of coffee beans if no control measures are applied [6]. CBB has been estimated to cause more than USD 500 million in annual losses worldwide [7], which is also why CBB is considered one of the most serious coffee pests [8]. Native to the African continent, *H. hampei* was detected in Mexico for the first time in 1978 in Soconusco, Chiapas, very close to the border with the Republic of Guatemala [9]. Currently, it is prevalent in various coffee-growing regions of the country [10].

CBB has been the focus of extensive research worldwide. One of the topics that has attracted considerable attention is how this pest is affected by microclimatic conditions. In particular, the relation between emergence and air temperature and humidity has been the subject of various studies [11,12]. Rain is reported to act as a “trigger” for the mass emergence of adult CBB females [13]. Emerged females then disperse and colonize the coffee berries of the following crop. Males, on the other hand, die without leaving their natal berry [7] because their atrophied wings impede flight. The mass emergence of CBB refers to the migration of the surviving population of flying females from the old coffee crop to the new one. In southern Mexico, Central America, and the Caribbean, rainfall decreases significantly after the end of the coffee harvest, and CBB adults tend to remain on residual or unharvested old coffee berries. Adults sheltering in unharvested berries enter reproductive diapause to avoid death due to lack of food and decreased humidity during the dry season [14,15,16]. In the coffee-growing region of Soconusco, the reproductive diapause of CBB is terminated by the sporadic rains that occur mainly between March and April (intercropping period) [14,15,16,17]. In response to the rain, females abandon, in large numbers, the berries that provided them shelter [15]. The mass emergence of CBB has been studied using traps baited with alcoholic attractants since the late 20th century [18,19]. However, there is a significant lack of knowledge regarding the influence of coffee plantation shade on the intensity and flight patterns of emerged adult females. More specifically, there is a lack of knowledge regarding the presence or absence of shade tree conditions in the microclimate for this kind of crop and the impact of these variables on the incidence of pests and diseases [20]. Information about the shade–microclimate–insect relationship for BCC is contradictory and reveals a complex problem [21]. For example, Barrera and Covarrubias [22] reported that the infestation of this pest increased in a coffee plantation with dense shade compared to one cultivated with less shade in Soconusco. However, Muñoz et al. [23] found the opposite in plantations around Lake Yojoa, Honduras: more infestation was observed in full sun than in shade. Likewise, Mariño et al. [24], in Puerto Rico, and Constantino et al. [25], in Colombia, reported more infestation in shaded than in sun-exposed plantations. Jaramillo et al. [26], in Kenya, reported that infestation with *H. hampei* was highest in plantations exposed to full sun. 

Conflicting reports in the published literature regarding CBB infestation of shaded and unshaded coffee plantations prompted two questions in our research group: (1) would shade trees also affect the intensity and patterns of adult female CBB mass emergence? To answer this question, we hypothesized that not only more CBB females would be captured in coffee grown under shade trees, but we also would expect to find differences in the seasonal and daily patterns of mass CBB emergence between shaded and unshaded plantations. (2) Which microclimatic variables would be associated with the rains that stimulate the mass emergence of this insect species during the intercropping period in coffee plantations? To address the second question, we hypothesized that microclimatic variables such as temperature and relative humidity, among others, would be associated with rain to cause the mass emergence of this insect. Answering these questions and testing these hypotheses would further deepen our knowledge about CBB. Also, this information is essential to determine the optimal period for the control of flying CBB females. Traps baited with alcoholic attractants are also useful to study the response of flying CBB females to environmental stimuli and to understand the dispersal and colonization processes of this pest in different coffee agroecosystems [17]. The objective of this research project was to study the effect of microclimate on the mass emergence of CBB in coffee plantations under shade as well as direct sun exposure. Accordingly, our research project aims to: (i) identify the most important microclimate factors that influence mass emergence, (ii) resolve the relationship between these microclimate factors and mass emergence, and (iii) determine the seasonal and diel patterns of mass emergence.

## 2. Materials and Methods

### 2.1. Study Site

The present research was carried out during the first half of 2022 on the Alianza farm, a 10–12-year-old Robusta coffee plantation (*Coffea canephora* Pierre ex A. Froehner) located in the municipality of Cacahoatán, Soconusco region, state of Chiapas, Mexico, at a mean altitude of 670 m above sea level and at the geographical coordinates of 15.046667 north latitude and −92.182778 west longitude (Figure 1). We selected this farm because it had a coffee plot grown under shade trees and another plot receiving full sunlight and infested with CBB. Furthermore, both plots were managed using the same agronomic practices and were located far enough apart from one another to avoid interference but close enough to avoid the effect of environmental variables unrelated to the present research.

The Soconusco region is located between 15°19′ north latitude and 92°44′ west longitude and covers an area of 4664 km^2^, which represents 6.2% of the area of the state of Chiapas [27]. It is the seventh largest region of the state and includes 15 municipalities, 11 of which (73.3%) produce coffee. The municipality of Cacahoatán occupies the third place in terms of planted hectares (9002.55 ha) and harvested volume (13,607.42 tons of coffee berries) at the state level (after the municipalities of Tapachula and Chilón) and the second place in the Soconusco region (after Tapachula) [28].

### 2.2. Experimental Plots

The study was carried out in two experimental plots of Robusta coffee, one grown under shade trees (*Tabebuia donnell-smithii* Rose and *Terminalia oblonga* [Ruiz and Pav.] Steud.) and the other grown in full sun exposure. Although the shade trees in the experimental plot are not frequently used as shade for coffee in Cacahoatán, this plot was chosen because it was located near a coffee plot with full sun exposure; this condition is not easy to find on the same farm in the region. At the site occupied by each trap in the shaded plot, canopy cover was recorded with the HabitApp app (https://play.google.com/store/apps/details?id=com.scrufster.habitapp&hl=en_US (accessed on 5 February 2024)) installed on an Android cell phone; The average percentage (±standard error) of shade in this plot was 70.33 ± 1.18% (*n* = 18).

The plots were located on flat terrain (<10 degrees) and approximately 300 m apart to avoid or reduce differences in rainfall intensity and patterns as well as other environmental factors. The rectangular plots (5 × 280 m) covered an area of 1400 m^2^. Each plot consisted of 420 coffee trees planted in three rows, with 2 m spacing within and 3 m between rows. The plots were managed using the same agronomic practices, and no pest control was carried out. At the beginning of the experiment, the percentage of CBB-perforated coffee fruits (% pf) in the shaded plot was lower than in the sun-exposed plot for fruits sampled (*n* = 30 per plot) from branches (shade: 0.0% pf vs. sun: 30.04 ± 8.11% pf) but similar for fruits sampled (*n* = 30 per plot) from the ground (shade: 24.05 ± 7.44% pf vs. sun: 24.78 ± 6.21% pf).

Because we wanted to gather the most representative CBB mass emergence data possible, no insecticides were applied at the beginning or during the experiment to affect the postharvest spatial dispersion of the insect population as little as possible and to guarantee a high number of flying females.

### 2.3. CBB Trapping

In each plot, 18 ECOIAPAR traps [15] were installed to capture adult CBB females. ECOIAPAR is a handmade trap made from disposable transparent plastic soft drink bottles (1.5 to 2.0 l capacity). A window was made in the central part of the bottles for the release of the attractant and as entry point for the flying CBB females. An attractant-containing diffuser was placed inside the bottle (at the middle part). The diffuser consisted of a 9.0 mL glass bottle with a rubber stopper that had a circular perforation of approximately 0.2 mm^2^. The attractant was composed of a methanol/ethanol (3:1) mixture, which has proven to be very effective in capturing flying CBB females under field conditions [29,30,31,32]. A previous study determined that the release rate of this trap was, on average (±standard error), 133.60 (±11.42) mg/day [33]. A water/propylene glycol mixture (8.5:1.5, 500–600 mL mixture; food-grade 99% propylene glycol, Glycol United States Pharmacopeia) was added to the bottom of the bottle to retain and preserve the captured CBB before removing them from the trap. Traps were suspended from the branches of coffee trees (approximately 1.5 m above ground level) and were spaced 14.0 m apart along a straight line. Variants of the ECOIAPAR trap have been used for the monitoring and control of flying CBB females during the intercropping period in several Latin American countries [29,30,31].

The traps were checked hourly from 08:00 h to 18:00 h to collect captured CBB females. This activity was carried out weekly on two successive days, using one day for each of the experimental plots. Captured females were placed in plastic bags by trap and time of day with a small amount of pure ethyl alcohol to prevent escape and were then transported to the laboratory for counting using a stereoscopic microscope at 6.7× to 45× total magnification (VE-S6, VELAB Mexico, Mexico City, Mexico). This procedure was repeated for 13 consecutive weeks from 24 February to 19 May 2022.

### 2.4. Microclimate Variables

Microclimate variables were recorded in the experimental plots every 5 min with a computerized meteorological station (Davis Instruments, Hayward, CA, USA) (Table 1) during the capture and collection of female CBB (08:00–18:00 h). Air temperature and relative humidity were calibrated using a data logger (USB-502-LCD, Measurement Computing Corporation, Norton, MA, USA).

### 2.5. Data Analysis

#### 2.5.1. Comparison of CBB Caught in Sun and Shade

The mean number of CBB caught for the 18 traps (number of CBB females/trap/hour/sampling date), or response variable, was compared between plot types or conditions (sun and shade), applying a regression model with negative binomial response [34]. A negative binomial model was used because it fit the data better than a Poisson model (overdispersion of data). Three regressor variables were used: sampling date (Sam_d) with 13 levels, time of day (Time) with 11 levels, and sun or shade condition of the plot (Condic) with 2 levels. These variables were adjusted as categorical variables. The total number of treatments in the model was 286 (13 × 11 × 2). The fitted model was:*η* = *β*_0_ + *β*_1×1_ + *β*_2×2_ + *β*_3×3_ + *β*_4×1×2_ + *β*_5×1×3_ + *β*_6×2×3_
(1)

where: *X*_1_ = *Sam_d*; *X*_2_ = *Time*; and *X*_3_ = *Condic*; *β*_1_… *β*_6_ were the parameters, and *η* was the link function of the regression model. The model fit was carried out with the maximum likelihood method using the R software version 4.3.2 [35], and multiple comparisons of the treatments were made using the 95% confidence intervals (95% CI) for the estimated response of each treatment.

#### 2.5.2. Effect of Microclimate on the Mass Emergence of CBB

The effect of microclimate variables on CBB capture was determined through a mixed model with negative binomial response [36] that had the following structure:*log*(*μ_ij_*) = *x′_ij_* *β* + *z′_ij_*
*β′*(2)
where *log*(*μ_ij_*) = logarithm of the mean number of caught CBB for 18 traps (number of CBB females/trap/hour/sampling date/plot type) (link function of the random factor); *x′_ij_* = time (trapping date and time), experimental plot condition (shaded and unshaded), wind speed, precipitation, maximum air temperature, equilibrium moisture content (EMC), dew point, ultraviolet index, and mean humidity (fixed factors); *z′_ij_* = traps per time-date per experimental plot (random factor); and *β* and *β′* = parameters. The model was fitted with the GlimmTBM package [37]; multiple comparisons were performed with the Effects package [38], and plots were drawn with the R software version 4.3.2 [35] ggplot2 package [39].

We chose a model with fixed effects because it allowed us to capture the individual characteristics of the study variables and control their impact on CBB mass emergence. The variables were considered fixed effects because we assumed that their effect on mass emergence was constant across all observations. This way, we were able to eliminate the influence of unobserved factors, which would allow us a more precise estimation of the effects of interest [40]. Microclimatic variables were selected based on published information, e.g., [7,11,12], and empirical data. The assumptions of the mixed model were verified with respect to the normality of the random factor.

## 3. Results

### 3.1. Seasonal and Daily Fluctuation in Microclimate Variables in Shade and Sun

Regarding seasonal fluctuation (sampling dates), it was observed that the variables related to illumination (solar radiation, Figure 2a; ultraviolet radiation, Figure 2b) and air temperature (Figure 2c) did not show any particular trend across the sampling dates, both in the sun-exposed as well as the shaded plot (*R*^2^ ≤ 0.0329). The remaining microclimate variables (related to humidity (Figure 2d–g) and wind (Figure 2h,i)), tended to increase slightly and in a linear fashion in both plot types as time progressed. Only the barometric pressure showed a different trend between the experimental plots (Figure 2j): while barometric pressure remained constant in the sun-exposed plot, a slight and linear downward trend was observed in the shaded plot as time progressed.

Studied microclimate variables exhibited diel fluctuation (time), except the number of wind speed samples, which did not show a definable trend in function of time for both plots (Figure 3h). The other variables fluctuated and exhibited convex or concave vertical curves (according to the Stalliviere-Corrêa classification [41]) across the hours of the day in both experimental plots. Most variables exhibited a convex curve: variables related to illumination (solar radiation, Figure 3a; ultraviolet radiation, Figure 3b), air temperature (Figure 3c), precipitation (Figure 3e), dew point (Figure 3f), wind speed (Figure 3i), and barometric pressure (Figure 3j). Their curve maxima are located between 9:00 and 10:00 h for the barometric pressure; between 11:00 and 12:00 h for solar radiation, ultraviolet radiation, and air temperature; and between 14:00 and 16:00 h for dew point, wind speed, and precipitation. The variables that exhibited a concave curve were relative humidity (Figure 3d), with a curve minimum at 12:00 h, and equilibrium moisture content (Figure 3g), with its minimum between 14:00 and 15:00 h.

The mean barometric pressure (761.00 ± 0.31 mmHg in sun, 760.25 ± 0.31 mmHg in shade) and the accumulated precipitation in 13 days (7.46 mm in sun, 8.67 mm in shade) were similar in both experimental plots. However, the coffee trees in the plot exposed to full sun received more solar radiation (419.83 ± 23.54 W/m^2^ in sun, 94.70 ± 8.09 W/m^2^ in shade) and ultraviolet radiation (1.05 ± 0.06 µW/cm^2^nm in sun, 0.07 ± 0.02 µW/cm^2^nm in shade) than the coffee trees in the shaded plot. 

Consequently, maximum air temperature (27.58 ± 0.26 °C in sun, 25.43 ± 0.23 °C in shade) and dew point (21.62 ± 0.28 °C in sun, 20.92 ± 0.40 °C in shade) were higher in the full sun plot than in the plot under tree shade. Mean relative humidity (71.55 ± 1.84% in sun, 76.92 ± 1.94% in shade) and equilibrium moisture content (13.38 ± 0.44% in sun, 16.04 ± 0.67% in shade) were lower in the sun-exposed than in the shaded plot. Analysis of the wind data revealed that both plots received the same number of wind speed samples (116.93 ± 0.04 samples), but wind speed was higher in the sun-exposed (1.42 ± 0.17 km/h) than in the shaded plot (0.21 ± 0.09 km/h).

### 3.2. Comparison of CBB Captured in Shade and Sun

The number of captured CBB females was adequately adjusted to the models we developed to measure the fixed effects of our study (Table 2 and Table 3). However, these models proved to inadequately estimate or forecast CBB mass emergence. Therefore, Figure 4 (seasonal capture) and Figure 5 (daily capture) can only be used to compare the effects of the studied variables on mass emergence but not to estimate the number of captured females. Thus, the following paragraphs emphasize the fluctuations and trends in the number of CBB captured generated by the model. Observed captures are reported when considered necessary.

In both the sun and shade plots, there were two peaks in the number of captures of CBB females during the period in which this research was carried out (Figure 4). More females were caught in the second (28 April to 12 May) than in the first capture peak (3 March to 7 April). The highest number of CBB captures occurred in the shade plot on 10 March. On 9–10 March and 16–17 March, when the highest number of captures was registered, the observed average number (±standard error) of CBB females/trap/day in shade and sun were 3588.72 (±629.80) and 1897.33 (±180.91), respectively.

The mixed model with negative binomial response (Equation (1), Table 2) applied to the mean number of captured CBB showed highly significant differences between dates, time of day, and plot condition (shade or sun). Furthermore, according to this model, highly significant interactions were recorded between dates and plot condition; dates and time of day; plot condition and time; and between dates, plot condition, and time (Table 2).

According to the 95% confidence limits, significantly more CBB females were captured in the shaded plot than in the sun-exposed plot on 8 (61.5%) of the 13 sampling dates (Figure 4). On the 13 sampling dates during the intercropping period, 1.8 times more flying CBB females were captured (117,522 observed specimens) in the shaded than in the full sun-exposed plot (66,459 observed specimens). Except for the sampling date of 17 March, the non-significant differences between shade and sun corresponded to sampling dates with a very low number of captures under both conditions (Figure 4). 

The largest number of CBB females was captured between 12:00 and 17:00 h (Figure 5). During this period, 97.0% and 78.6% of the total number of captured flying females were caught in the tree-shaded and the sun-exposed plot, respectively. Also, during this period, significantly more captures were recorded in the shaded than in the sun-exposed plot, except at 17:00 h, when the difference between both conditions was not significant. In the shaded plot, the largest number of CBB was captured between 14:00 to 16:00 h, whereas in the sun-exposed plot, the largest number was caught between 15:00 h and 17:00 (Figure 5). Between 14:00 and 16:00 h (when the largest number of captures was registered), the observed average number (±standard error) of CBB females/trap/hour varied from 127.18 (±21.34) to 160.11 (±21.36) in shade, and from 38.21 (±4.88) to 97.62 (±11.88) in sun.

### 3.3. Relationship between Microclimate Variables and the Mass Emergence of CBB

The mixed model with negative binomial response (Equation (2), Table 3) indicated that the log number of captured CBB females fitted a polynomial function of degree 3 [poly(Time 3)], both in the plot with full solar exposure as well as in the shaded plot. Except for the condition of the plots (Condic) and solar radiation (Sol_rad), and according to the model, microclimate variables were observed to be significantly related to the mass emergence of CBB (Table 3). Barometric pressure was not considered in the final model equation because its behavior was similar in both shade conditions.

Likewise, the model indicated significant interactions between the polynomial function of degree 3 with the sun and shade condition of the two plots [poly(Time, 3): Condic]; plot condition with the sampling date (Condic: Sam_d); and the sampling date with solar radiation (Sam_d: Sol_rad) (Table 3).

According to this model, the CBB mass emergence response was positive for the variables precipitation (Rain), dew point (Dew_p), and the number of wind speed samples received by the meteorological station (Samp_vi). On the contrary, CBB mass emergence response was negative for maximum air temperature (Max_aT), mean relative humidity (Mean_RH), ultraviolet radiation index (Uv_in), wind speed (Win_sp), and equilibrium moisture content (EMC) (Figure 6).

## 4. Discussion

The mass emergence triggered by sporadic rains during the dry period in Soconusco, particularly between March and April, occurs at a key moment for the survival and dispersal of CBB. During this period, thousands of female CBB leave the coffee berries of the previous crop, where they hatched and developed, and fly out in search of new berries from the following crop. Migration constitutes a pivotal phase in the CBB life cycle and population dynamics [14,15,16]. The offspring of CBB females that find a suitable berry for reproduction will contribute to the survival and multiplication of the species. But, also, by abandoning the berries that have served as their refuge, adult CBB females expose themselves to various mortality factors, including pest control methods used by some producers [7]. A wide variety of control strategies have been developed, including cultural control practices that make the habitat less suitable for CBB. These include weed management, pruning of coffee and shade trees and, above all, the collection of residual coffee fruits to eliminate shelters and food for the pest during the intercropping period [42]. Other strategies include ethological control techniques, which seek to capture flying females using alcoholic traps [43]; biological control methods, which involve the release of parasitoid insects [44] and the spraying of entomopathogenic fungi [45]; and chemical control techniques, which apply contact insecticides to kill flying females [46].

These control methods should be used as part of an integrated pest management (IPM) strategy to achieve better CBB control results [6,47,48]. It is therefore crucial to gain a better understanding of the effect of rainfall on the microclimate and, subsequently, of the impact on CBB mass emergence in plantations with different shade conditions. In this context, the presence or absence of shade trees has been reported to condition the microclimate of coffee plantations and this, in turn, has an impact on the incidence of pests and diseases [20].

Although our model was not able to adequately predict the number of CBB captures, it did allow us to analyze the effect of the studied variables on CBB mass emergence and answer the two questions we posed. Regarding the first question (would shade trees also affect the intensity and pattern of mass emergence of adult females of this pest?), we found that CBB mass emergence was clearly higher in the shaded coffee plot, which is in agreement with previous studies from the region [22]. However, the emergence pattern was similar in both plots. Regarding the second question (what microclimatic variables would be associated with the rains that stimulate the mass emergence of this insect species during the intercropping period in coffee plantations?), it was more difficult to predict which microclimatic variables affect mass emergence and to characterize the relationship of these variables to mass emergence—even though the literature tells us that CBB mass emergence is triggered by rain [13]. Our observations indicate a complex interaction between shade trees, microclimate, and mass emergence of CBB.

A first result of our research showed that there was a greater number of CBB females captured in the shaded plot (117,522 females in total) than in the plot exposed directly to the sun (66,459 females in total), a finding that is in agreement with previous reports from other authors [22,24,25]. However, the response of CBB mass emergence to microclimate was similar in both conditions, indicating that the presence or absence of shade trees does not modify the type of interaction between shade level, microclimate, and mass emergence for this pest.

Some interesting observations were made as regards the time of day at which CBB mass emergence was at its peak. In our experiments, the highest emergence of adult females was registered between 14:00 and 16:00 h in the shaded plot and one hour later (15:00 to 17:00 h) in the full sun-exposed plot. This result coincides with observations from Malaysia by Corbertt [49], who reported an emergence peak between 14:00 and 17:00 h, and from Nicaragua by Féliz et al. [50], who recorded a peak between 12:00 and 17:00 h. In our experiments, emergence occurred one or two hours earlier than reported from East Java (16:00 to 18:00 h), by Leefmans [51], and from the Philippines (16:00 to 17:00 h), by Morallo-Rejesus and Baldos [52]. Our research shows that CBB emergence rises after noon due to the values of some microclimate variables, which increase from the beginning of the day until reaching a maximum around noon and then gradually decrease again (convex curve). Examples of these variables are solar radiation, ultraviolet radiation, maximum air temperature, barometric pressure, precipitation, dew point, and wind speed. Other variables acted simultaneously but in opposite directions (convex curve), for instance, relative humidity and equilibrium moisture content. 

Our observations indicate a positive relation between female CBB mass emergence and precipitation, which coincides with a report from Hawaii by Johnson and Manoukis [7]. Unlike these authors, however, our results also indicated a negative relationship between emergence and air temperature, relative humidity, and wind speed. No relation was observed between solar radiation and emergence. Comparing our results with a study from Brazil by De Sousa et al. [53], we find several similarities but also differences. De Sousa et al. recognized a positive relationship between CBB flight and temperature, sunlight, and wind, whereas, in our case, this relationship was negative. However, like De Sousa et al. [53], our results indicate a negative relationship between CBB emergence and relative humidity. In our case, the negative relationship between CBB mass emergence and relative humidity occurred at a seasonal (emergence decreased and relative humidity increased as the sampling weeks progressed) as well as a daily level (mass emergence was lower before 13:00 h and after 17:00 h, when relative humidity was at its highest). Discrepancies between the results reported by Johnson and Manoukis [7] and De Souza et al. [53], on the one hand, and our study results, on the other hand, can be explained by differences in objectives, methodology, and environmental conditions, as Mariño et al. [24] suggested for studies of this type. Regarding the methodology, it is important to highlight that our research was different from others because it covered almost the entire intercropping period, since our interest was to study the mass emergence of CBB. This is significant, because mass emergence in this period occurs after the end of the reproductive diapause, as the first sporadic rains of the year arrive [14,15]. It has been shown that the dispersal behavior of CBB females in the intercropping period differs from that in other periods of the year. In the intercropping period, female CBB tend to fly higher [54], and it is assumed that by flying higher, the females want to take advantage of the wind currents above the canopy of the shade trees to spread.

We are convinced that studies such as the present one are needed to gain additional insight into the effect of microclimate on the emergence of CBB, especially during the period in which the mass emergence of this pest occurs. Also, since our experiments were performed during the first half of 2022 under typical La Niña climate conditions [55], it is necessary to conduct additional research to determine whether the interaction level of shade–microclimate–mass emergence of CBB would change under the influence of El Niño, a meteorological phenomenon whose conditions of lower humidity and higher air temperature favor the reproduction, dispersion, and infestation of this pest [56,57]. In Colombia, a country that does not experience extreme variations in average air temperatures during the year due to its proximity to the Earth’s equator, CBB is affected by the increase or decrease in air temperature and rainfall during El Niño or La Niña. For example, Constantino et al. [56] reported that increased drought and air temperature (>1–2 °C) during an El Niño event favored an increase in CBB populations and flights which, in turn, caused an increase in infestation levels and damage to coffee berries. In contrast, increased rainfall and decreased temperature during La Niña negatively affected CBB emergence and infestation [56]. In Columbia, according to Gil et al. [57], the temperature was higher than 21 °C, soil humidity was low (<30%), precipitation was less than 200 mm/month, a water deficit was experienced for one or two months, and the number of hours of sunshine was greater than 230 h/month during the dry periods caused by El Niño. Therefore, these authors assumed that humidity, air temperature, and shelter conditions in coffee plantations grown under shade trees during El Niño events will favor the survival, development, and multiplication of CBB [57]. If we consider the study results from Colombia and take into account that our experiments were conducted during a La Niña event, we can assume that, in our case, CBB mass emergence during an El Niño event might be even greater in shaded than in full sun-exposed coffee plantations.

Our results are relevant for various study areas, such as those that concentrate on modelling CBB development, reproduction, and dispersal in different situations. Such modelling studies require field data—like the data acquired by our research group—to feed the parameters or assumptions of their models [58]. Our study is also of interest to pest management practitioners and researchers, since knowledge of the interactions between CBB and microclimatic variables can function as a guide to determine the optimal moment (e.g., peaks of mass emergence) for the control of exposed flying females [7]. Coffee producers can eliminate flying females by mass trapping, the release of the adult CBB parasitoid *Phymastichus coffea* LaSalle (Hymenoptera: Eulophidae), the spraying of the entomopathogenic fungus *Beauveria bassiana* (Balsamo) Vuillemin (Hypocreales: Clavicipitaceae), or the use of synthetic contact insecticides [6,47,48]. As our findings, as well as results from De Sousa et al. [53] suggest, the best time to release *P. coffea*, spray *B. bassiana*, or apply chemical insecticides, is in the afternoon (after 12:00 h), when CBB females are the most exposed.

## 5. Conclusions

In conclusion, more flying females were captured in the shaded than in the sun-exposed coffee plot during CBB mass emergence in the intercropping period. In both shade conditions, the largest number of flying females were captured between 12:00 and 17:00 h. By abandoning their natal coffee berries at this period of the day, the CBB female population could be more exposed to natural and artificial mortality factors. Our research revealed that some microclimatic variables affect positively (dew point and number of wind speed samples received by the meteorological station) or negatively (maximum air temperature, average relative humidity, ultraviolet radiation, wind speed, and equilibrium moisture content) the seasonal and diel patterns of CBB mass emergence in association with rainfall. Monitoring some of these microclimate variables could help establish more precisely the optimal time for controlling flying CBB females during mass emergence in the intercropping period.

## Figures and Tables

**Figure 1 insects-15-00124-f001:**
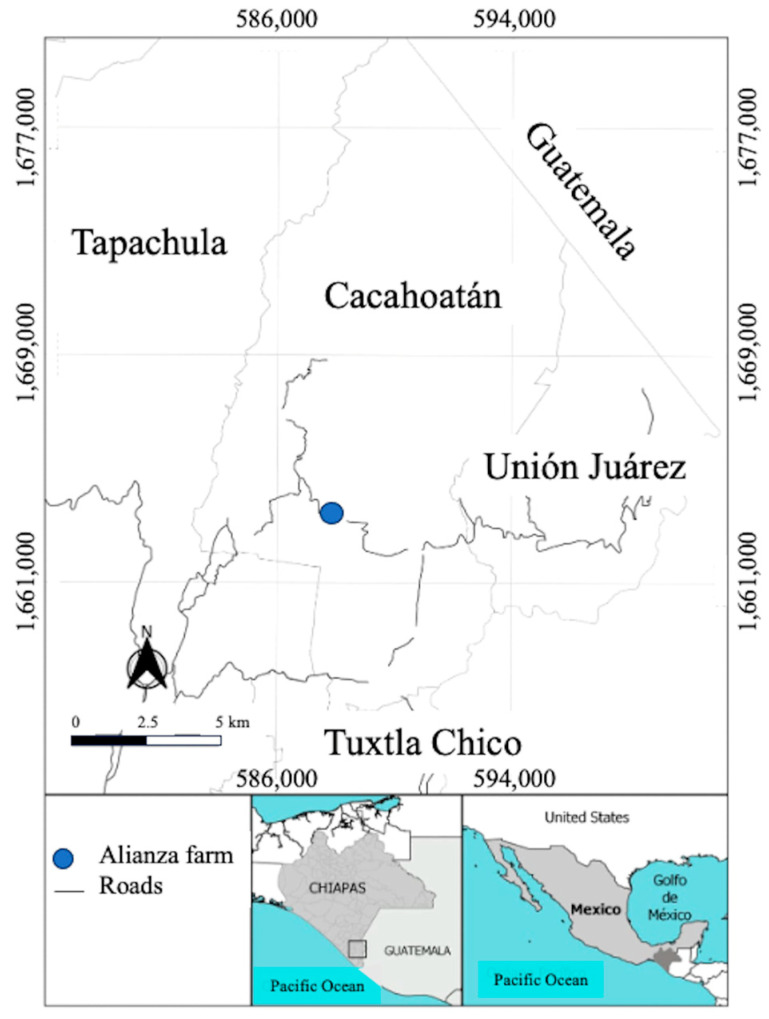
Location of the study area and the Robusta coffee (*Coffea canephora*) plantation (670 m above sea level) in the Soconusco region, Chiapas, Mexico.

**Figure 2 insects-15-00124-f002:**
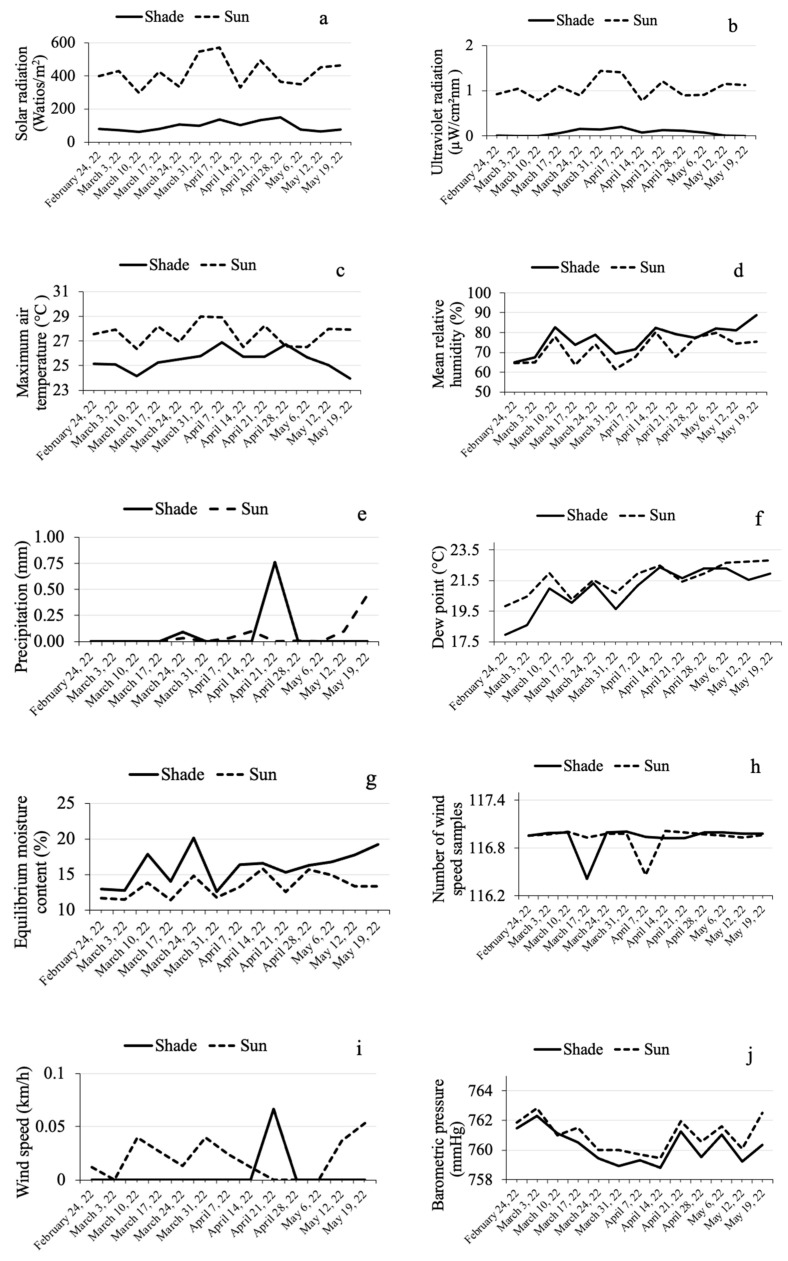
Seasonal (sampling dates) fluctuation in microclimate variables (day/date) recorded using a meteorological station in two plots of Robusta coffee (*Coffea canephora*), one exposed to full sun and the other under the shade of trees, during the CBB mass emergence period from 24 February to 19 May 2022. (**a**) Solar radiation/day/date; (**b**) ultraviolet radiation index/day/date; (**c**) maximum air temperature/day/date; (**d**) mean relative humidity/day/date; (**e**) precipitation/day/date; (**f**) dew point/day/date; (**g**) equilibrium moisture content/day/date; (**h**) wind speed samples/day/date; (**i**) wind speed/day/date; (**j**) barometric pressure/day/date.

**Figure 3 insects-15-00124-f003:**
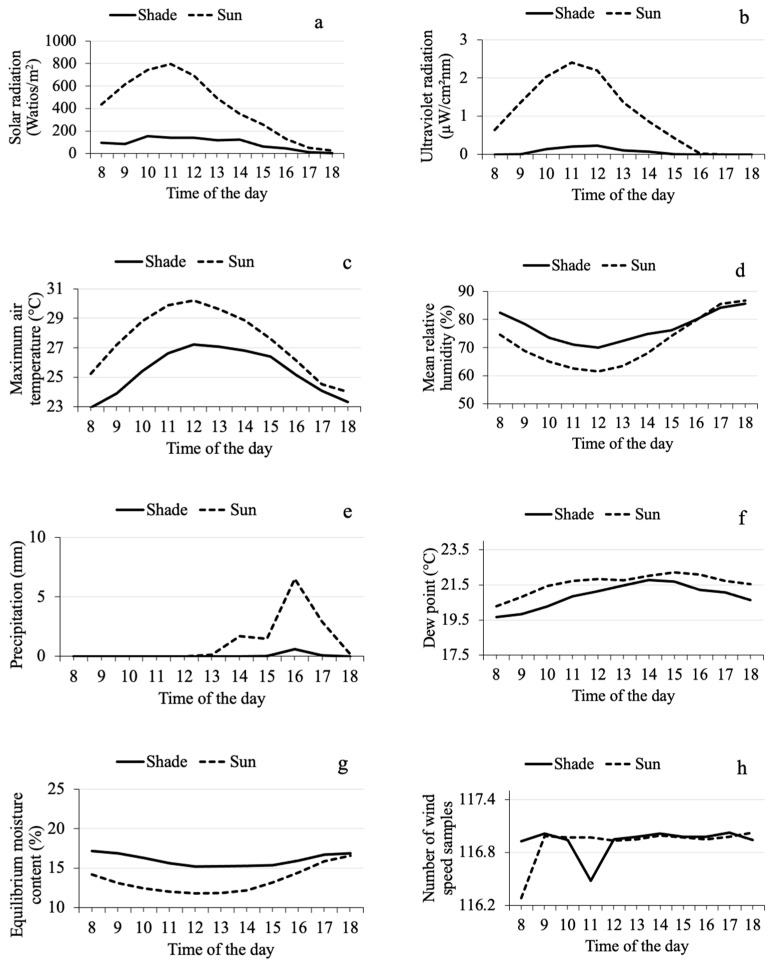
Diel (time of day) fluctuation in microclimate variables (hour/day) recorded using a meteorological station in two plots of Robusta coffee (*Coffea canephora*), one exposed to full sun and the other under the shade of trees, during the CBB mass emergence period from 24 February to 19 May 2022. (**a**) Solar radiation/hour/day; (**b**) ultraviolet radiation index/hour/day; (**c**) maximum air temperature/hour/day; (**d**) mean relative humidity/hour/day; (**e**) precipitation/hour/day; (**f**) dew point/hour/day; (**g**) equilibrium moisture content/hour/day; (**h**) wind speed samples/hour/day; (**i**) wind speed/hour/day; (**j**) barometric pressure/hour/day.

**Figure 4 insects-15-00124-f004:**
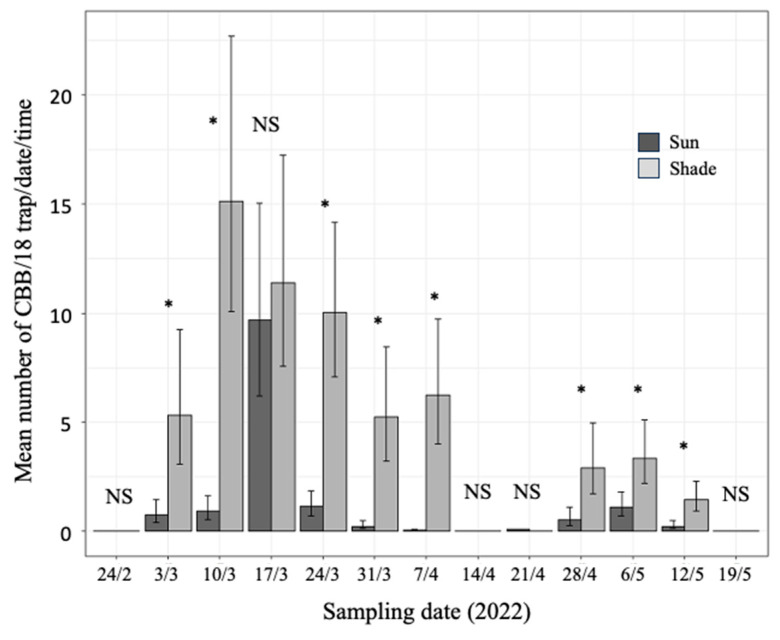
Mean number (±95% CI) of CBB females captured per sampling date (CBB/18 traps/date/time) in two plots of Robusta coffee (*Coffea canephora*), one exposed to full sun and the other under the shade of trees, during the mass CBB emergence from 24 February to 19 May 2022. * Difference between the plot exposed to full sun and the plot under the shade of trees, according to 95% CI (*, significant; NS, not significant).

**Figure 5 insects-15-00124-f005:**
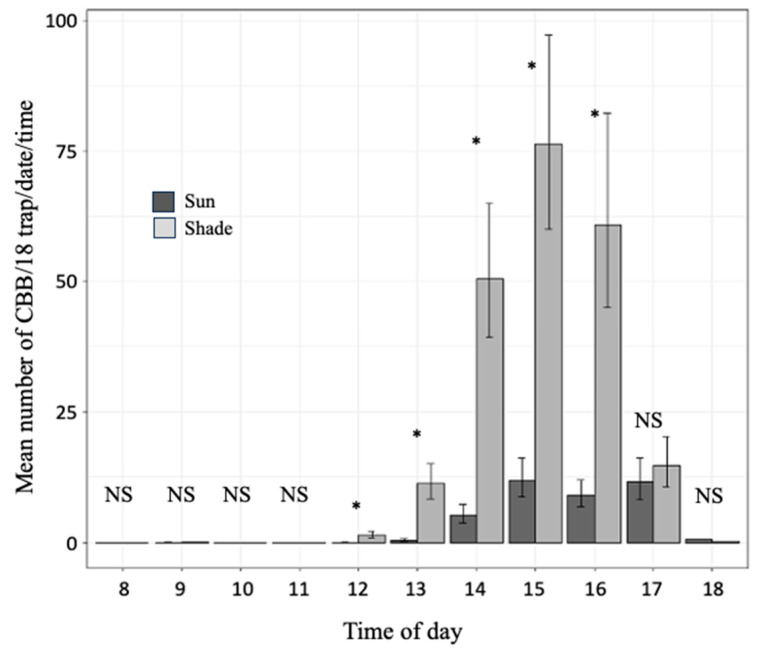
Mean number (±95% CI) of CBB females captured per time of day (CBB/18 traps/date/time) in two plots of Robusta coffee (*Coffea canephora*), one exposed to full sun and the other under the shade of trees, during the mass CBB emergency from 24 February to 19 May 2022. * Difference between the plot exposed to full sun and the plot under the shade of trees, according to 95% CI (*, significant; NS, not significant).

**Figure 6 insects-15-00124-f006:**
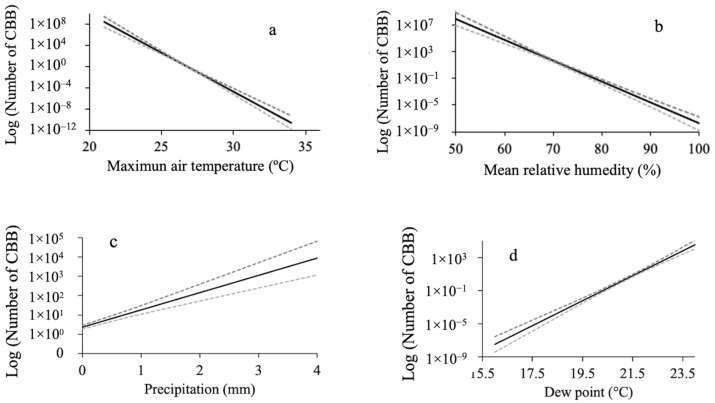
Relationship between logarithm of the mean number of captured CBB females (CBB/18 traps/date/time) and some variables of the microclimate in two plots of Robusta coffee (*Coffea canephora*), one exposed to full sun and the other under the shade of trees, during the CBB mass emergence period from 24 February to 19 May 2022. (**a**) Maximum air temperature (Max_aT); (**b**) relative humidity (Mean_RH); (**c**) precipitation (Rain); (**d**) dew point (Dew_p); (**e**) ultraviolet radiation (Uv_in); (**f**) wind speed (Win_sp); (**g**) number of wind speed samples received (Samp_vi); and (**h**) percentage of equilibrium moisture content (EMC). The dotted lines represent 95% confidence intervals.

**Table 1 insects-15-00124-t001:** Microclimate variables recorded in the experimental plots of Robusta coffee (*Coffea canephora*) with a computerized climatological station.

Variable	Key	Units
Solar radiation	Sol_rad	W/m^2^
Wind speed	Win_sp	km/h
Precipitation	Rain	mm
Ultraviolet radiation index	Uv_in	µW/cm^2^nm
Mean relative humidity	Mean_RH	%
Maximum air temperature	Max_aT	°C
Dew point	Dew_p	°C
Equilibrium moisture content	EMC	%
Barometric pressure	Bar	mmHg
Wind speed samples	Samp_vi	Number

**Table 2 insects-15-00124-t002:** Deviation analysis (type II tests) of the regression model with negative binomial response (Equation (1)) applied to the mean number of captured CBB females (CBB/18 traps/date/time) in two plots of Robusta coffee (*Coffea canephora*), one exposed to full sun and the other under the shade of trees, during the CBB mass emergence period from 24 February to 19 May 2022.

Source of Variation	Logistic Regression Model Chi-Square	Degree of Fredom	*p*
Sampling date (Sam_d)	5062.5	12	<0.0001
Plot condition, sun/shade (Condic)	1254.4	1	<0.0001
Time of day (Time)	14,454.1	10	<0.0001
Sam_d: Condic	916.7	12	<0.0001
Sam_d: Time	2994.8	120	<0.0001
Condic: Time	788.9	10	<0.0001
Sam_d: Condic: Time	1535.8	109	<0.0001

**Table 3 insects-15-00124-t003:** Deviance analysis (Wald type II chi-square tests) of the mixed model with negative binomial response (Equation (2)) applied to the logarithm of mean number of captured CBB females (CBB/18 traps/date/time) in two plots of Robusta coffee (*Coffea canephora*), one exposed to full sun and the other under the shade of trees, during the CBB mass emergence period from 24 February to 19 May 2022.

Sources of Variation	Chi-Square	Degree of Freedom	*p*
Polynomial function of degree 3 (poly(*Time*, 3))	2102.3146	3	<0.0001
Plot condition, sun/shade (Condic)	3.7927	1	0.0515
Sampling date (Sam_d)	1628.8696	12	<0.0001
Solar radiation (Sol_rad)	1.4951	1	0.2214
Wind speed (Win_sp)	41.7858	1	<0.0001
Precipitation (Rain)	60.9070	1	<0.0001
Ultraviolet radiation index (Uv_in)	10.7842	1	0.0010
Mean relative humidity (Mean_RH)	238.1158	1	<0.0001
Maximum air temperature (Max_aT)	268.9199	1	<0.0001
Dew point (Dew_p)	270.6297	1	<0.0001
Equilibrium moisture content (EMC)	8.0771	1	0.0045
Wind speed samples (Samp_vi)	23.1626	1	<0.0001
poly(Time, 3): Condic	190.4641	3	<0.0001
Condic: Sam_d	945.3627	12	<0.0001
Sam_d: Sol_rad	321.9021	12	<0.0001

## Data Availability

The data presented in this study are available on request from the corresponding author.

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
