# Peer review of "Effect of Microclimate on the Mass Emergence of Hypothenemus hampei in Coffee Grown under Shade of Trees and in Full Sun Exposure"

_insects, 2024, doi:10.3390/insects15020124_

Round 1
Reviewer 1 Report
Comments and Suggestions for Authors
Reviewer 2 Report
Comments and Suggestions for Authors
Title:
ok.
Abstract
Integrate a concise sentence summarizing the key implications of the findings.
Introduction
Clarity and Flow:
The introduction is comprehensive, but consider breaking down the long sentences for better readability (e.g., lines 64-71).
In line 92, consider rephrasing: "Based on the differences reported in the literature on CBB infestation between coffee plantations with or without shade..."
Objective Clarity:
Clearly state the research questions as separate items for better clarity (lines 94-97).
Grammar and Style:
Consider revising line 94 for conciseness: "In this investigation, we explore whether shade affects the intensity and pattern of mass emergence of adult female CBB."
Materials and Methods
Site Description:
Consider providing a brief rationale for selecting the "Alianza" farm in the Soconusco region for the study (lines 110-113).
In Figure 1 caption (lines 115-116), mention the elevation of the study area.
Experimental Design:
Clearly state the reasons for choosing the specific shade tree species in the study (lines 127-130).
Trapping Details:
Clarify the rationale for choosing the specific attractant mixture of methanol: ethanol (3:1) for the CBB traps (lines 137-139).
Data Collection:
Specify if there was any standardization or calibration performed for the meteorological station recording microclimate variables (lines 148-150).
Data Analysis:
In the "Comparison of CBB catches in sun and shade" section (lines 155-162), provide a brief rationale for choosing the negative binomial regression model.
In the "Effect of microclimate on the mass emergence of CBB" section (lines 169-179), specify the basis for selecting the fixed factors and their expected impact on CBB emergence.
Statistical Methods:
Clearly state if the model assumptions were checked, especially for the mixed model (lines 172-179).
Results
Presented Well
Discussion:
Introduction to Key Findings:
Consider opening the discussion with a concise recap of the key findings to set the context for subsequent details (lines 315-325).
Comparison with Previous Studies:
Clearly articulate differences in results compared to previous studies (lines 352-357).
Explain how variations in objectives, methodologies, and environmental conditions might contribute to discrepancies (lines 358-361).
Consideration of Climatic Events:
Expand on the potential influence of climatic events like La Niña or El Niño on shade-microclimate-mass emergence interactions (lines 363-365).
Relevance of Results:
Emphasize the relevance of the study results for various applications, such as modeling, pest management, and consideration for different climatic events (lines 366-372).
Practical Implications:
Provide practical insights for coffee producers, highlighting specific control measures based on the identified interactions (lines 372-375).
General comments
After carefully reviewing the manuscript, I must commend the author for their skillful writing and overall presentation. However, I have identified several areas where the manuscript could be improved. These suggestions will help the author further enhance the manuscript's readability, structure, and impact.
Comments on the Quality of English LanguageCareful proofreading is required.
Reviewer 3 Report
Comments and Suggestions for Authors
This manuscript is addressing a relevant topic related to the emergence of coffee berry borer (CBB) from infested berries and its association with weather conditions. Results are very interesting and show potential implications in the management of this insect pest. All comments, questions are given for the improvement of this manuscript. See the report/ PDF file attached.
